# Evaluation of health-related quality of life using EQ-5D in China during the COVID-19 pandemic

Weiwei Ping[1]*, Jianzhong Zheng[1], Xiaohong Niu[2], Chongzheng Guo[1], Jinfang Zhang[2], Hui Yang[1], Yan Shi[1]

**1** Department of Preventive Medicine, Changzhi Medical College, Changzhi, Shanxi, P.R.China, **2** Heji Hospital Affiliated to Changzhi Medical College, Changzhi, Shanxi, P.R.China

\* weiweip@czmc.edu.cn

**Data Availability Statement:** All relevant data are within the paper and its Supporting Information files.

**Funding:** The study was funded by the Soft Science Project of Shanxi of China (2017041040-

## Abstract

### Objective

Since December 2019, an increasing number of cases of the 2019 novel coronavirus disease (COVID-19) infected by severe acute respiratory syndrome coronavirus 2 (SARS-CoV-2) have been identified in Wuhan, Hubei Province, China. Now, more cases have been reported in 200 other countries and regions. The pandemic disease not only affects physical health who suffered it, but also affects the mental health of the general population. This study aims to know about the impact of the COVID-19 epidemic on the health-related quality of life (HRQOL) of living using EQ-5D in general population in China.

### Methods

An online-based survey was developed and participants were recruited via social media. The questionnaires included demographic and socioeconomic data, health status, the condition epidemic situation and EQ-5D scale. The relationships of all factors and the scores of EQ-5D were analyzed. Logistic regression model were used to the five health dimensions.

### Results

The respondents obtained a mean EQ-5D index score of 0.949 and a mean VAS score of 85.52.The most frequently reported problem were pain/discomfort (19.0%) and anxiety/depression (17.6%). Logistic regression models showed that the risk of pain/discomfort and anxiety/depression among people with aging, with chronic disease, lower income, epidemic effects, worry about get COVID-19 raised significantly.

### Conclusion

The article provides important evidence on HRQOL during the COVID-19 pandemic. The risk of pain/discomfort and anxiety/depression in general population in China raised significantly with aging, with chronic disease, lower income, epidemic effects, worried about get

4), the Soft Science Project of Shanxi of China (2018041033-1) to WP, and by the prevention and control COVID-19 project of Changzhi Medical College by XN (XG202004).

**Competing interests:** The authors have declared that no competing interests exist.

COVID-19 during the COVID-19 pandemic. The results from each categorical data can be used for future healthcare measures among general population.

## Introduction

Since December 2019, an increasing number of cases of the 2019 novel coronavirus disease (COVID-19) infected by severe acute respiratory syndrome coronavirus 2 (SARS-CoV-2) have been identified in Wuhan, Hubei Province, China. Until April 9, 2020, the rapid spread of the virus had caused 83249 cases and 3344 deaths in China [1,2].More cases have been reported in 200 other countries and regions, including the USA, Italy, Spain, France, Germany and so on [3]. As a consequence, since the beginning of the COVID-19 epidemic in China, to minimize the risk of infection, the Chinese government, health agency and medias recommended people decreased go out and travel, wear a face mask at outside and wash hands frequently after outside by mobile phone short note, TV, We-chat and community education [4].The Chinese government began to provide social distancing advice minimizing the risk of the virus. It is necessary to know about the impact of the COVID-19 epidemic on the health-related quality of life (HRQOL) of living in China.

In recent years, health-related quality of life (HRQOL) has pay worldwide attention, and several multidimensional health status classifications have been increasingly used to describe and evaluate HRQOL in China [5,6]. Healthy China 2030 that is an outline for "the Healthy China 2030" initiative, has been announced in 2016, aims to promote life expectancy and improve HRQOL in all Chinese people [7].

Some scales have been widely developed to measure HRQOL. Generic instrument of HRQOL, for example: the World Health Organization Quality of Life (WHOQOL—BREF) [8], can be used to compare HRQOL across various diseases/conditions and can be use for different populations to assess the impact of various interventions on QOL [9]. Condition-specific instrument of HRQOL (e.g.EORTCQLQ2-C30), can only be used to compare HRQOL specific diseases/conditions for specific populations (e.g. cancer or diabetes) [10]. On the other hand, the quality of life should be measured on a utility scale on which 1.0 corresponds to full health and 0.0 corresponds to death. Therefore, HRQOL scales can be classified into psychometric instrument (e.g. SF-36) and utility instrument (e.g. the Health Utilities Index) [11,12]. The EQ-5D is a simple but widely used instrument based on characteristic that can be used to measure generic population based on utility [13]. Internationally, studies using the EuroQol (EQ-5D) survey have demonstrated lower scores in older individuals compared with younger [14,15], lower scores in women than in men, lower scores in individuals of lower socioeconomic status compared with of higher socioeconomic status [16,17].

EQ-5D has been translated into Chinese, its validity and reliability validity were evaluated by [18, 19], and the EQ-5D instrument is a valid measure for Chinese HRQOL. Its population value set has been development using a time trade-off (TTO) approach in 2014 [13] and VAS approach in 2015 [20].Recently, some studies have been used the instrument to measure HRQOL in China [21].We conducted questionnaires using EQ-5D to evaluate Health-Related Quality of Life during the COVID-19 pandemic in Changzhi city, Shanxi province, China.

## Methods

### Respondents

Changzhi city is a small city located in east-south of Shanxi province in north China. Changzhi city covers an area of 13864km$^2$, with a population of 3.468 million people. During the period

of COVID-19 epidemic, there were 8 definite case and 42 suspected cases in the Changzhi city. The survey was conducted from March 2 to March 10 after receiving ethical approval from the Ethics Committee of Changzhi Medical College. An online-based survey was developed and participants were recruited via social media (e.g. We-chat). A statement about informed consent was included with the questionnaire, and returning the questionnaire was considered to constitute provision of informed consent. At March 10, 1500 questionnaire were returned by respondents. Of these, 215 respondents were not residents that lived in Changzhi city according to the location.146 were deemed unusable due to using time is less than 100 seconds. The left 1139 were deemed usable.

## Survey questionnaire

The survey questionnaires included the following information.

1. The demographic and socioeconomic data of the respondents. The demographic variables included age and sex, we categorized age into six groups (<18, 18–29, 30–39, 40–49, 50–59, and 60+ years).The socioeconomic variables included marital status, employment status, educational level and income level in the local. The categorization of these variables was showed in Table 1.

2. Health status: The health status variables included chronic condition and Health-related behaviors. A chronic condition was defined as a chronic condition by a doctor, for whom either the symptoms persisted or relevant medical treatment continued over the past six months. Chronic conditions covered 12 major medical chronic conditions: hypertension, heart disease (including coronary heart disease and other heart condition), stroke, hyperlipidemia, liver disease, diabetes mellitus and other endocrine disease, respiratory disease, urinary and reproductive disease, musculoskeletal disease, gastrointestinal disease, dermal diseases, and dental caries or other dental diseases. Respondents answered "yes" if they had one or more chronic condition. Respondents were classified by the number of chronic disease as no, with one chronic disease, with two chronic diseases, with three or more chronic disease. Health-related behaviors included regular exercise(moderate or vigorous exercise for >30min,≧3times/week), sufficient sleep(7-8h/day).

3. The condition epidemic situation: worry about got COVID-19 (respondents were classified by 5 degree); whether influenced by pandemic at the aspects of social activity, usual activity, sleeping, diet, and exercise. Respondents answered "yes" if they had one or more aspects effects. Respondents were classified by the number of chronic disease as no and yes regardless of affected on one or more aspects.

4. EQ-5D scale: the Chinese version of EQ-5D was included in the questionnaire. It is a self-completed instrument for describing and valuing quality of health states defined by the EQ-5D index. It measures five dimensions of health: mobility, self-care, usual activities, pain/discomfort, and anxiety/depression, as well as overall health rated on a VAS. Each dimension has three levels, corresponding to "no problem", "some problem", and "extreme problem", allowing for $3^5$ (i.e., 243) possible health combinations. The VAS scores ranged from 0(worst health) to (best health). HRQOL results measured by the EQ-5D were converted to an index score using the China value set which ranges from -0.149 to 1.00 [13]. A negative value represents a health status worse than being dead, 0 represents being dead and 1 represents the state of full health.

**Table 1. Characteristics of respondents and EQ-5D index and visual analogue (VAS) scores.**

| | EQ-5D-3L Index | | | | EQ-5D-3L VAS | |
|---|---|---|---|---|---|---|
| | N | (%) | mean(SD) | p value | mean(SD) | p value |
| Total | 1139 | 100 | 0.949(0.102) | | 85.52(19.373) | |
| Sex | | | | | | |
| Male | 460 | 40.4 | 0.947 (0.108) | 0.482 | 86.89(18.459) | 0.050 |
| Female | 679 | 59.6 | 0.951 (0.098) | | 84.60(19.930) | |
| Age (year) | | | | | | |
| <18 | 36 | 3.2 | 0.963 (0.074) | 0.001 | 95.94(5.560) | 0.047 |
| 18–29 | 271 | 23.8 | 0.975 (0.063) | | 85.12(23.034) | |
| 30–39 | 322 | 28.3 | 0.963 (0.090) | | 85.45(19.622) | |
| 40–49 | 276 | 24.2 | 0.953 (0.084) | | 85.20(18.377) | |
| 50–59 | 158 | 13.9 | 0.898 (0.150) | | 85.35(14.899) | |
| 60+ | 76 | 6.7 | 0.889 (0.141) | | 83.84(19.070) | |
| Marital status | | | | | | |
| Married | 869 | 76.2 | 0.967 (0.107) | 0.130 | 85.63(17.848) | 0.005 |
| Unmarried | 233 | 20.5 | 0.962 (0.081) | | 85.64(22.180) | |
| Divorced/widowed | 38 | 3.3 | 0.944 (0.087) | | 75.47(29.784) | |
| Employment status | | | | | | |
| Employed | 693 | 60.8 | 0.957 (0.088) | 0.001 | 84.48(20.527) | 0.018 |
| Retired | 106 | 9.3 | 0.886 (0.152) | | 84.38(17.501) | |
| Unemployed | 340 | 29.9 | 0.954 (0.102) | | 88.01(19.373) | |
| Chronic disease condition | | | | | | |
| No chronic disease | 671 | 58.9 | 0.979 (0.053) | 0.001 | 89.44(17.184) | 0.001 |
| With one chronic disease | 248 | 21.8 | 0.936 (0.112) | | 81.44(19.802) | |
| With two chronic disease | 112 | 9.8 | 0.916 (0.101) | | 81.57(20.683) | |
| With three or more chronic disease | 108 | 9.5 | 0.828 (0.175) | | 74.63(22.864) | |
| Education level | | | | | | |
| Primary school and Below | 203 | 17.8 | 0.948(0.085) | 0.803 | 84.73(20.466) | 0.752 |
| Junior middle school | 256 | 22.5 | 0.946(0.109) | | 85.56(21.021) | |
| Senior middle school | 346 | 30.4 | 0.954(0.098) | | 86.39(18.326) | |
| University and above | 334 | 29.3 | 0.944(0.112) | | 84.77(18.802) | |
| Family income(in the local) | | | | | | |
| Low | 58 | 5.1 | 0.945(0.133) | 0.148 | 81.10(11.274) | 0.006 |
| Lower | 614 | 53.9 | 0.951(0.099) | | 86.67(18.209) | |
| Middle | 318 | 27.9 | 0.952(0.106) | | 83.66(21.571) | |
| Higher | 90 | 7.9 | 0.925(0.091) | | 81.20(21.389) | |
| High | 59 | 5.2 | 0.964(0.072) | | 84.75(19.907) | |
| Worry about got COVID-19 | | | | | | |
| Very high | 32 | 2.8 | 0.868(0.220) | 0.001 | 82.19(19.144) | 0.001 |
| High | 102 | 9.0 | 0.918(0.124) | | 73.63(25.020) | |
| Low | 484 | 42.5 | 0.948(0.089) | | 85.66(17.074) | |
| Very low | 521 | 45.7 | 0.962(0.095) | | 87.93(19.311) | |
| Epidemic effects | | | | | | |
| Yes | 660 | 57.9 | 0.936(0.116) | 0.001 | 83.21(21.439) | 0.001 |
| No | 479 | 42.1 | 0.968(0.074) | | 88.71(15.567) | |

SD: standard deviation. *p*-value: *p* values come from *t* test or ANOVA *or* Mann-Whitney U test or Kruskall-Wallis test.

## Data analysis

Statistical analyses were carried out using the Statistical Product and Service Solutions (SPSS) sofeware23.0. Mean and standard deviations were calculated for continuous variables, frequencies and percentages for categorical variables. The relationships of all factors and the scores of EQ-5D were analyzed with t-test, analysis of variance (ANOVA), and nonparametric statistics (Mann-Whitney U test or Kruskall-Wallis test). The percentage of people in each dimension was calculated and $x^2$-test were performed to examine the statistical significance of the difference between groups in the percentage of reported problems. Fisher's exact test was used when exact theory frequency less than 1. Logistic regression model were used to the five health dimensions as dependent variables (0 = no problem, 1 = some/extreme problem). Statistical significance was set at 0.05 using two-side tests.

The permission of the study was obtained from the Ethics Committee of Changzhi Medical College on March, 2020.

# Results

## Characteristics of respondents

The respondents had a mean age of 38.3 years (SD: 12.5 years; range12-78years). 40.4% were men and all of them were Han ethnicity. The mean years of formal education were 11.5 years. 60.8%were employed by full time, 41.1% respondents reported diagnosed with one or more chronic diseases over the past six months. 11.8% worried got the COVID-19, and 57.9% respondents had been affected by epidemic disease on social activity, usual activity, sleeping, diet, and exercise on one or more field (Table 1).

## EQ-5D results

The respondents obtained a mean EQ-5D index score of 0.949 (SD: 0.102) and a mean VAS score of 85.52(19.37). Highest possible EQ-5D index score reported at 71.9% respondents, highest possible VAS score at 24.2%. Older age (p<0.001), Unemployed (p<0.001), with chronic disease (p<0.001), low family income (p<0.05), worry about got COVID-19 (p<0.001), and have epidemic effects (p<0.001) were associated with lower EQ-5D index score. The VAS score obtained consistent results as those of EQ-5D index score.

The most frequently reported problem were pain/discomfort (19.0%), followed by anxiety/depression (17.6%), self-care (1.1%) was the least frequently reported problem. Men were more likely to report problem in mobility (6.1%) than women (2.4%). Compared with less 18 years respondents, other age respondents were more likely to report problem in five dimension of EQ-5D, and above 60 years group respondents reported the most problem in mobility (13.2%), usual activities (7.9%), pain/discomfort (52.6%), and anxiety/depression (23.7%). Unemployed respondents reported the most problem in self-care (1.9%), usual activities (11.3%), pain/discomfort(49.1%), and anxiety/depression (26.4%) than employed respondents. Compared with no chronic disease respondents, with chronic disease (one or more) respondents were more likely to report problem in five dimension of EQ-5D.Those who very worry about got COVID-19 were more likely to report problem in mobility (12.5%), self-care (6.3%), usual activities(18.7%), pain/discomfort (43.7%), and anxiety/depression (37.5%) than those who no worry. Those who reported had been affected by epidemic reported the more problem in mobility (5.2%), usual activities (3.0%), and anxiety/depression (22.4%) than those who reported had not been effected by epidemic (Table 2).

**Table 2. Percentage of reported any problem in 5 dimensions of EQ-5D.**

| | Mobility | | | Self-care | | | Usual Activities | | | Pain/discomfort | | | Anxiety/depression | | |
|---|---|---|---|---|---|---|---|---|---|---|---|---|---|---|---|
| | No | Some or extreme | *p* value | No | Some or extreme | *p* value | No | Some or extreme | *p* value | No | Some or extreme | *p* value | No %/N | Some or extreme | *p* value |
| Total | 96.1 | 3.9 | | 98.9 | 1.1 | | 98.1 | 1.9 | | 81.0 | 19.0 | | 82.4 | 17.6 | |
| Sex | | | | | | | | | | | | | | | |
| Male | 93.9 | 6.1 | **0.001** | 98.3 | 1.7 | 0.062 | 97.4 | 2.6 | 0.172 | 81.7 | 18.3 | 0.685 | 83.0 | 17.0 | 0.660 |
| Female | 97.6 | 2.4 | | 99.4 | 0.6 | | 98.5 | 1.5 | | 80.6 | 19.4 | | 82.0 | 18.0 | |
| Age(yea) | | | | | | | | | | | | | | | |
| <18 | 100.0 | 0.0 | **0.001** | 94.4 | 5.6 | **0.001**△ | 100.0 | 0.0 | **0.001**△ | 100 | 0.0 | **0.001** | 77.8 | 22.2 | **0.008** |
| 18–29 | 98.5 | 1.5 | | 99.3 | 0.7 | | 100.0 | 0.0 | | 95.6 | 4.4 | | 87.5 | 12.5 | |
| 30–39 | 98.1 | 1.9 | | 99.4 | 0.6 | | 100.0 | 0.0 | | 87.0 | 13.0 | | 85.1 | 14.9 | |
| 40–49 | 97.1 | 2.9 | | 100.0 | 0.0 | | 98.6 | 1.4 | | 79.7 | 20.3 | | 81.2 | 18.8 | |
| 50–59 | 89.9 | 10.1 | | 96.2 | 3.8 | | 92.4 | 7.6 | | 58.2 | 41.8 | | 74.7 | 25.3 | |
| 60+ | 86.8 | 13.2 | | 100.0 | 0.0 | | 92.1 | 7.9 | | 47.4 | 52.6 | | 76.3 | 23.7 | |
| Marital status | | | | | | | | | | | | | | | |
| Unmarried | 97.4 | 2.6 | 0.203 | 98.3 | 1.7 | 0.530 | 100.0 | 0.0 | **0.020** | 92.3 | 7.7 | 0.001 | 83.7 | 16.3 | 0.065 |
| Married | 95.6 | 4.4 | | 99.1 | 0.9 | | 97.5 | 2.5 | | 77.9 | 22.1 | | 82.7 | 17.3 | |
| Divorced/widowed | 100.0 | 0.0 | | 100.0 | 0.0 | | 100.0 | 0.0 | | 84.2 | 15.8 | | 68.4 | 31.6 | |
| Employment status | | | | | | | | | | | | | | | |
| Employed | 97.1 | 2.9 | **0.001** | 99.4 | 0.6 | 0.145 | 99.1 | 0.9 | **0.001** | 83.8 | 16.2 | **0.001** | 84.1 | 15.9 | **0.007** |
| Retired | 97.1 | 2.9 | | 98.2 | 1.8 | | 98.8 | 1.2 | | 84.7 | 15.3 | | 81.8 | 18.2 | |
| Unemployed | 86.8 | 13.2 | | 98.1 | 1.9 | | 88.7 | 11.3 | | 50.9 | 49.1 | | 73.6 | 26.4 | |
| Chronic disease condition | | | | | | | | | | | | | | | |
| No chronic disease | 99.4 | 0.6 | **0.001** | 99.7 | 0.3 | **0.001** | 100.0 | 0.0 | **0.001** | 93.4 | 6.6 | **0.001** | 89.9 | 10.1 | **0.001** |
| With one chronic disease | 94.4 | 5.6 | | 98.4 | 1.6 | | 97.6 | 2.4 | | 75.8 | 24.2 | | 79.8 | 20.2 | |
| With two chronic disease | 96.4 | 3.6 | | 100.0 | 0.0 | | 96.4 | 3.6 | | 62.5 | 37.5 | | 71.4 | 28.6 | |
| With three and more chronic disease | 79.6 | 20.4 | | 94.4 | 5.6 | | 88.9 | 11.1 | | 35.2 | 64.8 | | 53.7 | 46.3 | |
| Education level | | | | | | | | | | | | | | | |
| Primary school and Below | 98.0 | 2.0 | **0.031** | 99.0 | 1.0 | 0.719 | 99.0 | 1.0 | **0.002** | 79.3 | 20.7 | 0.495 | 81.3 | 18.7 | 0.719 |
| Junior middle school | 93.8 | 6.2 | | 98.4 | 1.6 | | 95.3 | 4.7 | | 82.0 | 18.0 | | 82.0 | 18.0 | |
| Senior middle school | 97.7 | 2.3 | | 98.8 | 1.2 | | 98.3 | 1.7 | | 83.2 | 16.8 | | 81.5 | 18.5 | |
| University and above | 95.2 | 4.8 | | 99.4 | 0.6 | | 99.4 | 0.6 | | 79.0 | 21.0 | | 84.4 | 15.6 | |
| Family income level(in the local) | | | | | | | | | | | | | | | |
| Low | 93.1 | 6.9 | 0.066 | 100.0 | 0.0 | 0.377△ | 96.6 | 3.4 | 0.596 | 79.3 | 20.7 | **0.002** | 86.2 | 13.8 | **0.003** |
| Lower | 96.7 | 3.3 | | 99.0 | 1.0 | | 98.4 | 1.6 | | 79.8 | 20.2 | | 84.7 | 15.3 | |
| Middle | 96.9 | 3.1 | | 98.7 | 1.3 | | 97.5 | 2.5 | | 85.5 | 14.5 | | 80.5 | 19.5 | |
| Higher | 91.1 | 8.9 | | 100.0 | 0.0 | | 97.8 | 2.2 | | 68.9 | 31.1 | | 68.9 | 31.1 | |
| High | 96.6 | 3.4 | | 96.6 | 3.4 | | 100.0 | 0.0 | | 89.8 | 10.2 | | 86.4 | 13.6 | |
| Worry about get COVID-19 | | | | | | | | | | | | | | | |
| Very high | 87.5 | 12.5 | **0.001** | 93.8 | 6.2 | **0.019** | 81.3 | 18.7 | **0.001** | 56.3 | 43.7 | **0.001** | 62.5 | 37.5 | **0.001** |
| High | 90.2 | 9.8 | | 100.0 | 0.0 | | 98.0 | 2.0 | | 68.6 | 31.4 | | 72.5 | 27.5 | |
| Low | 97.5 | 2.5 | | 98.8 | 1.2 | | 98.3 | 1.7 | | 78.1 | 21.9 | | 80.2 | 19.8 | |

(*Continued*)

**Table 2.** (Continued)

| | Mobility | | | Self-care | | | Usual Activities | | | Pain/discomfort | | | Anxiety/depression | | |
|---|---|---|---|---|---|---|---|---|---|---|---|---|---|---|---|
| | No | Some or extreme | p value | No | Some or extreme | p value | No | Some or extreme | p value | No | Some or extreme | p value | No %/N | Some or extreme | p value |
| Very low | 96.5 | 3.5 | | 99.2 | 0.8 | | 98.8 | 1.2 | | 87.7 | 12.3 | | 87.7 | 12.3 | |
| Epidemic effects | | | | | | | | | | | | | | | |
| Yes | 94.8 | 5.2 | **0.008** | 99.2 | 0.8 | 0.538 | 97.0 | 3.0 | **0.002** | 76.1 | 23.9 | **0.001** | 77.6 | 22.4 | **0.001** |
| no | 97.9 | 2.1 | | 98.8 | 1.2 | | 99.6 | 0.4 | | 87.9 | 22.1 | | 89.1 | 10.9 | |

△: *p* values is the probability of Fish's exact test. Bold values are statistically significant.

## Logistic regression analysis

Each dimension of EQ-5D have been dichotomized, and have been used dependent variable. Sex, age, marital status, employment status, chronic disease condition, education level, family income level in the local, worry about get COVID-19, epidemic effects have been included as independent variables, multivariate logistic regression models were conducted, only those variables which exerted a significant relationship with any dimension from EQ-5D were reported in Table 3. The results showed that sex (OR = 2.621, 95%CI:1.333–5.150), age (OR = 2.028, 95%CI:1.458–2.820), marital (OR = 0.172,95%CI:0.052–0.570), with chronic disease (OR = 2.428, 95%CI:1.790–3.293), and epidemic effects(OR = 2.451, 95%CI:1.133–5.306) showed a significant relationship in mobility dimension; marital (OR = 0.166, 95%CI:0.039–0.707),

**Table 3. Multivariate logistic regression analysis results on the relationships between 5 dimensions of EQ-5D and influence factors.**

| Dimensions of EQ-5D | Influence factors | B | SE | p value | Odds ratio | 95%CI |
|---|---|---|---|---|---|---|
| Mobility | Sex | 0.963 | 0.345 | 0.005 | 2.621 | 1.333–5.150 |
| | Age | 0.707 | 0.168 | 0.001 | 2.028 | 1.458–2.820 |
| | Marital | -1.758 | 0.610 | 0.004 | 0.172 | 0.052–0.570 |
| | With chronic disease | 0.887 | 0.156 | 0.001 | 2.428 | 1.790–3.293 |
| | Epidemic effects | 0.897 | 0.394 | 0.023 | 2.451 | 1.133–5.306 |
| Self-care | Marital | -1.798 | 0.740 | 0.015 | 0.166 | 0.039–0.707 |
| | Employ | 0.675 | 0.333 | 0.043 | 1.964 | 1.022–3.773 |
| | With chronic disease | 1.167 | 0.299 | 0.000 | 3.212 | 1.787–5.774 |
| Usual Activities | Sex | 0.888 | 0.508 | 0.080 | 2.431 | 0.898–6.578 |
| | Age | 1.000 | 0.238 | 0.001 | 2.719 | 1.706–4.333 |
| | Employ | 0.734 | 0.352 | 0.037 | 2.084 | 1.044–4.158 |
| | With chronic disease | 0.861 | 0.247 | 0.001 | 2.365 | 1.459–3.835 |
| | Worry about get COVID-19 | -0.720 | 0.283 | 0.011 | 0.487 | 0.280–0.847 |
| | Epidemic effects | 2.214 | 0.815 | 0.007 | 9.156 | 1.852–45.276 |
| Pain/discomfort | Age | 0.780 | 0.091 | 0.001 | 2.182 | 1.825–2.610 |
| | Marital | -0.720 | 0.288 | 0.012 | 0.487 | 0.277–0.855 |
| | With chronic disease | 0.838 | 0.087 | 0.001 | 2.312 | 1.951–2.740 |
| | Worry about get COVID-19 | -0.416 | 0.121 | 0.001 | 0.660 | 0.520–0.836 |
| | Epidemic effects | 0.865 | 0.199 | 0.001 | 2.375 | 1.607–3.511 |
| Anxiety/depression | With chronic disease | 0.611 | 0.075 | 0.001 | 1.843 | 1.590–2.137 |
| | Family income | 0.231 | 0.087 | 0.008 | 1.259 | 1.061–1.494 |
| | Worry about get COVID-19 | -0.322 | 0.105 | 0.002 | 0.724 | 0.590–0.889 |
| | Epidemic effects | 0.731 | 0.181 | 0.001 | 2.076 | 1.455–2.963 |

employ(OR = 0.166, 95%CI:0.039–0.707) and with chronic disease(OR = 1.964, 95%CI:1.022–3.773) showed a significant relationship in self-care dimension; age(OR = 2.719, 95%CI:1.706–4.333), employ(OR = 2.084, 95%CI:1.044–4.158), with chronic disease (OR = 2.365, 95%CI:1.459–3.835), worry about got COVID-19 (OR = 0.487, 95%CI:0.280–0.847), and epidemic effects(OR = 9.156, 95%CI:1.825–45.276) showed a significant relationship in usual activities dimension. Age(OR = 2.182,95%CI:1.825–2.610), marital(OR = 0.487, 95%CI:0.277–0.855), with chronic disease (OR = 2.312, 95%CI:1.951–2.740),worry about got COVID-19 (OR = 1.201, 95%CI:0.520–0.836), and epidemic effects (OR = 2.375,95%CI:1.607–3.511) showed a significant relationship in pain/discomfort dimension. With chronic disease (OR = 1.843, 95% CI: 1.590–2.137), family income level in the local (OR = 1.259, 95%CI: 1.061–1.494), worry about got COVID-19(OR = 0.724, 95%CI: 0.590–0.889), and epidemic effects (OR = 2.076, 95%CI: 1.455–2.963) showed a significant relationship in anxiety/depression dimension.

## Discussion

The present study aimed to assess the HRQOL in Chinese population during the COVID-19 pandemic using the EQ-5D scale, the mean score for EQ-5D score and VAS scale were 0.949 (0.102) and 85.52(19.373) respectively. The results similar to the Singapore population score (0.95) that measured in general population used the UK time trade-off values [21], but higher than the score reported in the USA (0.87) in 2010 [22], Demark (0.889) in 2009 [23], Sri Lanka (0.85) in 2014 [24], and Japan (0.877) in 2011 [25]. The mean VAS score of our study (85.52) is higher than the national average (80.12) of in Heilongjiang of China and Taiwan population score (74.5) [26, 27]. The results revealed that the HRQOL in Chinese population during the COVID-19 pandemic perhaps had not been changed on the whole. As noted by the Sun et al [20], there are several possible reasons why the Chinese population obtained the higher score. Firstly, people in different countries may refer to the levels of health differently due to cultural differences. Secondly, the health status as well as the age and sex structure is different across countries. Thirdly, at the Changzhi city, the epidemic condition that only 8 definite cases had not affected largely life for mostly people.

Our study showed that the mean EQ-5D index score decreased with increasing age. Older people were more likely to reported problem in all of the five domains than younger people. Those who are of 50 and older dropped to a level of below average. The association between age and EQ-5D index score remained significant even after adjusting for all socio-demographic variables. The study showed that with chronic disease is the most significant variable, the most reported problems in all of the five domains, and the mean EQ-5D index score decreased rapidly with increasing the number of chronic disease. Aging had been a great challenge in China, 65 years and older has reached 12.5% in China in 2019 [28]. So, more people lived with disease for a long time, in particular, chronic non-communicable disease (NCD). Some study using EQ-5D also reported that QOL was lower among individuals with diabetes, gastrointestinal disease, hypertension, heart disease, and so on [29,30]. Our study showed that our respondents with three and above chronic diseases reported lower EQ-5D scores than other respondents, it similes to the Japan and Heilongjiang population results [25, 26]. The finding noted that aging people with three and above chronic diseases have lower QOL score in the period of epidemic, they should be pay close attention at the time.

Pain/discomfort was the most frequently reported problem in our study. The finding was consistent with the EQ-5D population studies from other countries. The proportion of our population reported pain/discomfort is similar to from Japan and Heilongjiang population results [25, 26], but lower than those reported from UK [31], Poland [32], Greece [33] and USA [22]. The bivariate analysis found that people who aging, unemployed, with one or more

chronic disease, very worry about get COVID-19, with epidemic effects were more likely to reported problem in pain/discomfort domain. The logistic regression analysis showed similar trend in pain/discomfort domain. Based on earlier experiences of SARS, MERS [34,35] and the limited recent evidence about get COVID-19 [36,37], pain/discomfort was associated with older age, low educational level, clinical severity, depression, anxiety, low quality of life. It is important to realize about pain/discomfort in the population during the COVID-19 pandemic.

Anxiety/depression was the second frequently reported problem that its proportion is close to pain/discomfort. The bivariate analysis and logistic regression analysis showed with chronic disease, lower income, worry about got COVID-19, have epidemic effects were more likely reported lower scores in anxiety/depression domain. The result was lower than in China [26], Japan [25], and Singapore [21]. Liu's [34] study about SARS in 2003 showed that perceived SARS-related risk level during the outbreak increased the odds of having a high level of depressive symptoms 3 years later. Kang 's study about COVID-19 in 2020 in Wuhan in China showed that medical and nursing staff had subthreshold mental health disturbances [36]. The number of people suffering from mental health impacts after a major event is often greater than the number of people who are physically injured, and mental health effects may last longer, therefore, it is necessary to pay attention to mental health, in particular, with chronic disease and lower income population.

## Limitation

This study has several limitations. First, compared with face-to-face interviews, online-based self-reporting survey has certain limitations. Second, in this study, the overall ceiling effects of the EQ-5D index may occur when measuring the quality of life of Chinese sample. Third, it is a cross-sectional study that conducted survey from March2 to March 10 that the period that epidemic has weakened in China, changes in QOL dropped off with the extension of time. A randomized prospective study could better determine correlation and causation. Fourth, the epidemic condition affects all of Chinese people; it now appears that it affects even the entire world. China is so large and diverse in culture and social development, and suffered varying degrees effects during of epidemic in all of area, a larger sample size coming from different areas in China is needed to verify the results. Lastly, the measure on worry of COVID-19 developed by ourselves in our study, but psychological distress instruments on COVID-19 have been developed by Ahorsu and Taylor in the recent study [38,39]. So, it is necessary to use appropriate instruments on COVID-19 (e.g., the Fear of COVID-19 Scale and the COVID Stress Scales) to measure mental health in the future study.

## Conclusion

The article provides important evidence on HRQOL during the COVID-19 pandemic. The risk of pain/discomfort and anxiety/depression in general population in China raised significantly with aging, with chronic disease, lower income, epidemic effects, worried about get COVID-19 during the COVID-19 pandemic. The results came from each categorical data can be used for future healthcare measures among general population.

## Supporting information

**S1 File. Questionnaires in Chinese.**
(DOC)

**S2 File. Questionnaires in English.**
(DOCX)

**S1 Data.**
(SAV)

## Acknowledgments

We thank all the participants who gave their time to make this project a reality.

## Author Contributions

**Conceptualization:** Jianzhong Zheng, Xiaohong Niu.

**Funding acquisition:** Jianzhong Zheng.

**Investigation:** Weiwei Ping, Chongzheng Guo, Jinfang Zhang, Hui Yang, Yan Shi.

**Methodology:** Weiwei Ping.

**Writing – original draft:** Weiwei Ping, Jinfang Zhang.

**Writing – review & editing:** Weiwei Ping.

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
