## [Decision Letter · Decision Letter 0]

13 May 2020

PONE-D-20-12772

Evaluation of Health-Related Quality of Life Using EQ-5D in China During the COVID-19 Pandemic

PLOS ONE

Dear Mrs ping,

Thank you for submitting your manuscript to PLOS ONE. After careful consideration, we feel that it has merit but does not fully meet PLOS ONE’s publication criteria as it currently stands. Therefore, we invite you to submit a revised version of the manuscript that addresses the points raised during the review process.

We would appreciate receiving your revised manuscript by Jun 27 2020 11:59PM. To enhance the reproducibility of your results, we recommend that if applicable you deposit your laboratory protocols in protocols.io, where a protocol can be assigned its own identifier (DOI) such that it can be cited independently in the future. For instructions see: http://journals.plos.org/plosone/s/submission-guidelines#loc-laboratory-protocols

We look forward to receiving your revised manuscript.

Kind regards,

Amir H. Pakpour, Ph.D.

Academic Editor

PLOS ONE

Journal Requirements:

Additional Editor Comments (if provided):

Reviewers' comments:

Reviewer's Responses to Questions

**Comments to the Author**

1. Is the manuscript technically sound, and do the data support the conclusions?

Reviewer #1: Partly

2. Has the statistical analysis been performed appropriately and rigorously? 

Reviewer #1: No

3. Have the authors made all data underlying the findings in their manuscript fully available?

Reviewer #1: Yes

4. Is the manuscript presented in an intelligible fashion and written in standard English?

Reviewer #1: Yes

5. Review Comments to the Author

Reviewer #1: I believe that the study entitled “Evaluation of Health-Related Quality of Life Using EQ-5D in China during the COVID-19 Pandemic” may bring something to the literature and help healthcare providers in caring general population. However, several parts of the manuscript need to be improved. Please see my specific comments below.

1. In Introduction, the sentence “The EuroQol (EQ-5D) is perhaps the most commonly used by researchers” is too strong. Please tone down the statement. If the authors want to keep this statement, please provide evidence. For example, how many papers have been using EQ-5D and how many papers have been using other quality of life instruments?

2. As the authors want to study in quality of life, they should firstly introduce different types of quality of life measures. From the aspect of “condition”, quality of life instruments can be classified into “condition-specific instrument” and “generic instrument”. From the aspect of “psychometrics”, quality of life instruments can be classified into “psychometric instrument” and “utility instrument”. The authors should clearly introduce these different types and clearly let the readers know that the EQ-5D is a generic instrument based on utility. Please refer to the following.

*References on “condition”

Lin, C.-Y., Lee, T.-Y., Sun, Z.-J., Yang, Y.-C., Wu, J.-S., & Ou, H.-t. (2017). Development of diabetes-specific quality of life module to be in conjunction with the World Health Organization Quality of Life Scale Brief Version (WHOQOL-BREF). Health & Quality of Life Outcomes, 15, 167.

Lin, C.-Y., Hwang, J.-S., Wang, W.-C., Lai, W.-W., Su, W.-C., Wu, T.-Y., Yao, G., & Wang, J.-D. (2019). Psychometric evaluation of the WHOQOL-BREF, Taiwan version, across five kinds of Taiwanese cancer survivors: Rasch analysis and confirmatory factor analysis. Journal of the Formosan Medical Association, 118(1), 215-222.

Lin, C.-Y. (2018). Comparing quality of life instruments: Sizing Them Up versus PedsQL and Kid-KINDL. Social Health & Behavior, 1, 42-47.

Pakpour, A. H., Chen, C.-Y., Lin, C.-Y., Strong, C., Tsai, M.-C., & Lin, Y.-C. (2019). The relationship between children's overweight and quality of life: A comparison of Sizing Me Up, PedsQL, and Kid-KINDL. International Journal of Clinical and Health Psychology, 19(1), 49-56.

*Reference on “psychometrics”

Lin, H. W., Li, C. I., Lin, F. J., Chang, J. Y., Gau, C. S., Luo, N., Pickard, A. S., Ramos Goñi, J. M., Tang, C. H., & Hsu, C. N. (2018). Valuation of the EQ-5D-5L in Taiwan. PloS One, 13(12), e0209344.

Lee, H. Y., Hung, M. C., Hu, F. C., Chang, Y. Y., Hsieh, C. L., & Wang, J. D. (2013). Estimating quality weights for EQ-5D (EuroQol-5 dimensions) health states with the time trade-off method in Taiwan. Journal of the Formosan Medical Association, 112(11), 699-706.

3. Regarding the 1500 surveys, how did the authors perform the random selection? Please provide the process. For example, did the authors obtain a list before sending out the surveys? If yes, how and where did the authors obtain the list? How was the representativeness of the list? Did the list include all the residents in Changzhi city?

4. In Results, the subheading of Regression analysis is misleading. Specifically, regression analysis gives an impression of “linear regression”. Therefore, the authors should explicitly mention “logistic regression” for the subheading.

5. Table 3. The heading of “CI 95 (Exp B)” is confusing. Please remove (Exp B). Also, it is unclear why there is a * sign before 1.798 for the marital variable. The title of Table 3 should clearly state whether this is multivariate logistic regression model or univariate logistic regression model.

6. Following the prior comment, it is unclear whether the authors conducted multivariate logistic regression or univariate logistic regression. This needs to be clear.

7. As the authors conducted a series of related analyses, they should adjust the p-values.

8. The authors should mention a limitation that, apart from the EQ-5D, all other measures used in the present study did not have been tested for psychometric properties. In particular, the measure on worry of COVID-19 seems to be developed by the authors themselves. Thus, the authors should acknowledge in the present study that there are available psychological distress instruments on COVID-19. Indeed, the authors found that anxiety/depression is the second frequently reported problem. Encouraging future studies using appropriate instruments on COVID-19 (e.g., the Fear of COVID-19 Scale and the COVID Stress Scales) is needed. Please see and cite the following references for discussion.

Ahorsu, D. K., Lin, C. Y., Imani, V., Saffari, M., Griffiths, M. D., & Pakpour, A. H. (2020). The Fear of COVID-19 Scale: Development and initial validation. International Journal of Mental Health and Addiction. Advance online publication. doi: 10.1007/s11469-020-00270-8.

Sakib, N., Mamun, M. A., Bhuiyan, A. K. M. I., Hossain, S., Mamun, F. A., Hosen, I., … Pakpour, A. H. (2020). Psychometric validation of the Bangla Fear of COVID-19 Scale: Confirmatory factor analysis and Rasch analysis. International Journal of Mental Health and Addiction. Advance online publication. doi: 10.1007/s11469-020-00289-x.

Satici, B., Gocet-Tekin, E., Deniz, M. E., & Satici, S. A. (2020). Adaptation of the Fear of COVID-19 Scale: Its association with psychological distress and life satisfaction in Turkey. International Journal of Mental Health Addiction. Advance online publication. doi: 10.1007/s11469-020-00294-0.

Soraci, P., Ferrari, A., Abbiati, F.A., Del Fante, E., De Pace, R., Urso A. Griffiths, M.D. (2020). Validation and psychometric evaluation of the Italian version of the Fear of COVID-19 Scale. International Journal of Mental Health and Addiction. Advance online publication. doi: 10.1007/s11469-020-00277-1.

Taylor, S., Landry, C., Paluszek, M., Fergus, T. A., Mckay, D., Asmundson, G. J. G. (2020). Development and initial validation of the COVID Stress Scales. Journal of Anxiety Disorders. Advance online publication. doi: 10.1016/j.janxdis.2020.102232.

9. Lastly, I think that the authors cannot claim to study the “impact of COVID-19 epidemic on quality of life” throughout the manuscript. The present study is a cross-sectional study and cannot provide such causality statement. Please revise all such statements.

6. PLOS authors have the option to publish the peer review history of their article (what does this mean?). If published, this will include your full peer review and any attached files.

Reviewer #1: No

---

## [Decision Letter · Decision Letter 1]

4 Jun 2020

Evaluation of Health-Related Quality of Life Using EQ-5D in China During the COVID-19 Pandemic

PONE-D-20-12772R1

Dear Dr. ping,

We’re pleased to inform you that your manuscript has been judged scientifically suitable for publication and will be formally accepted for publication once it meets all outstanding technical requirements.

Kind regards,

Amir H. Pakpour, Ph.D.

Academic Editor

PLOS ONE

Additional Editor Comments (optional):

Reviewers' comments:

Reviewer's Responses to Questions

**Comments to the Author**

1. If the authors have adequately addressed your comments raised in a previous round of review and you feel that this manuscript is now acceptable for publication, you may indicate that here to bypass the “Comments to the Author” section, enter your conflict of interest statement in the “Confidential to Editor” section, and submit your "Accept" recommendation.

Reviewer #1: All comments have been addressed

2. Is the manuscript technically sound, and do the data support the conclusions?

Reviewer #1: Yes

3. Has the statistical analysis been performed appropriately and rigorously? 

Reviewer #1: Yes

4. Have the authors made all data underlying the findings in their manuscript fully available?

Reviewer #1: Yes

5. Is the manuscript presented in an intelligible fashion and written in standard English?

Reviewer #1: Yes

6. Review Comments to the Author

Reviewer #1: The authors have satisfactory responded to my prior comments. I have no more comments and am glad to recommend publication in the present form. Congrats!

7. PLOS authors have the option to publish the peer review history of their article (what does this mean?). If published, this will include your full peer review and any attached files.

Reviewer #1: No

---

## [Editor Report · Acceptance letter]

10 Jun 2020

PONE-D-20-12772R1 

Evaluation of Health-Related Quality of Life Using EQ-5D in China During The COVID-19 Pandemic 

Dear Dr. Ping:

I'm pleased to inform you that your manuscript has been deemed suitable for publication in PLOS ONE. Congratulations! Your manuscript is now with our production department. 

Kind regards, 

on behalf of

Dr. Amir H. Pakpour 

Academic Editor

PLOS ONE